# The behavioral driving mechanism of ecological co-management among multiple subjects from the perspective of social network embedding: Evidence from coffee-producing areas in China

Xiumei Xu[1]*, Dengke Wang[2]

**1** School of Economics and Management, Baoshan University, Baoshan, China, **2** School of Business Administration, China University of Petroleum-Beijing at Karamay, Karamay, China

* 812994739@qq.com

## Abstract

The coffee-growing areas in the Gaoligong Mountains of China face ecological challenges including soil erosion and water pollution from traditional processing methods. To analyze the drivers of stakeholder participation in addressing these issues, this study integrates the Theory of Planned Behavior (TPB) and Social Network Embeddedness Theory (SNET), which together explain how individual cognitions and social structures shape cooperative behavior. Data from a stratified survey of 137 stakeholders were analyzed using PLS-SEM. The results demonstrate that (1) Both emotional networks (ENW) and suggestive networks (SNW) have significant direct effects on participation intention. (2) Indirectly, ENW enhances intention by strengthening individual behavioral attitude, subjective norm, and perceived behavioral control through emotional bonds and identity. (3) SNW improves intention primarily by boosting perceived behavioral control through information dissemination. And (4) the core TPB constructs (attitude, norm, control) are confirmed as key mediators. This paper contributes a validated integrated framework that elucidates the social-psychological pathways for fostering effective ecological co-management.

## Introduction

As global ecological and environmental issues become increasingly severe, ecological governance has become a focal point of attention for governments and academia worldwide [1]. Traditional unilateral approaches to governance have been empirically demonstrated to possess limited efficacy in resolving complex ecological challenges [2]. In contrast, the co-management paradigm, characterized by the involvement of multiple stakeholders, has progressively emerged as the predominant model in the field of ecological governance [3]. This model emphasizes the collective participation of various stakeholders, including government [4], businesses, communities and the public [5], to achieve ecological governance objectives through collaborative efforts.

**Data availability statement:** Original data of this paper are provided on FigShare DOI: https://figshare.com/articles/dataset/_/31169137 DOI: https://figshare.com/articles/dataset/_/31169143 DOI: https://figshare.com/s/0c159ab80cbfd6144ef4.

**Funding:** This work is supported by the Yunnan Province Philosophy and Social Science Planning (No. ZX2024ZD10).

**Competing interests:** The authors have declared that no competing interests exist.

It must be noted, however, that participatory behaviors within these multi-stakeholder frameworks are influenced by a multitude of factors, thereby rendering the effective stimulation and regulation of such participation a persistent challenge in ongoing scholarly inquiry [6].

The coffee cultivation base in the Lujiangba area of Gaoligong Mountain, Yunnan Province, serves as a typical case study for ecological governance. As a representative mountainous coffee-producing region, this area faces prominent local ecological challenges resulting from long-term intensive cultivation. First, slope topography and soil erosion during coffee cultivation have led to declining soil fertility and degradation of arable land quality. Second, the processing of coffee cherries remains heavily reliant on traditional washing methods. Due to the absence of advanced wastewater treatment facilities, this practice exerts considerable pressure on the local ecosystem. Third, the monoculture structure of coffee cultivation has significantly reduced local biodiversity, weakening ecosystem stability and resilience. Additionally, waste generated during coffee cherry processing should also be classified as a non-point source pollutant. Consequently, pollution arising from coffee cultivation and processing cannot be overlooked. Ecological governance is closely linked to the livelihoods of local farmers, making collaborative ecological management an urgent necessity. The effectiveness of governance in this region directly determines the sustainability of the local ecosystem and community well-being.

Current governance patterns in this region predominantly consist of unilateral initiatives by individual stakeholders, with truly collaborative multi-stakeholder approaches yet to be systematically implemented. The absence of effective coordination among diverse participants has consequently emerged as a primary constraint on regional ecological sustainability. Therefore, examining the participation intentions of various stakeholders in ecological co-management and their determining factors carries substantial academic and practical significance.

Grounded in the Theory of Planned Behavior (TPB) [7] and Social Network Embeddedness Theory (SNET) [8], this investigation develops an integrated analytical framework to explicate the behavioral mechanisms underlying stakeholder participation in ecological co-management. According to TPB, behavioral intention serves as the most proximate determinant of actual conduct, with such intention being shaped by behavioral attitudes, subjective norms, and perceived behavioral control [9].

Simultaneously, SNET posits that an individual's structural position within social networks exerts significant influence on decision-making processes [10]. In view of this, the study further introduces two embeddedness variables, emotional network (ENW) and suggestive network (SNW), to reveal the mechanisms through which social networks influence stakeholder participation intention in ecological co-management. The ENW reflects the emotional connections and satisfaction experienced by individuals, which can enhance their intrinsic motivation and commitment to ecological behaviors [11], while the SNW denotes the impact of information and suggestions obtained through social interactions on decision-making.

This paper primarily examines how ENW and SNW influence participation intentions among stakeholders at the Lujiangba coffee plantation, with particular focus on three investigative dimensions: (1) How do ENW and SNW directly impact stakeholders' individual behavioral attitudes (IBA), subjective norms (SN) and perceived behavioral control (PBC)? (2) How do ENW and SNW directly influence the behavioral intentions of stakeholders? (3) How do ENW and SNW indirectly influence the behavioral intentions of stakeholders through IBA, SN and PBC?

By addressing these questions, the study not only extends the theoretical applications of TPB and SNET in ecological governance research but also provides empirical foundations for policy formulation. The analytical methodology employs structural equation modeling (SEM), a multivariate technique capable of simultaneously evaluating complex relationships among multiple variables [12]. Data collection was conducted through a theoretically-grounded questionnaire.

The subsequent sections of this paper are organized as follows: Section 2 reviews relevant literature on TPB and SNET; Section 3 delineates the hypotheses and theoretical model. Section 4 details the research methodology; Section 5 presents the empirical findings; and Section 6 concludes with theoretical implications and policy recommendations.

## Literature review

### Application of the TPB in environmental participation

The TPB posits that behavioral intention is the most proximal determinant of actual behavior, and this intention is formed by three core constructs: attitude toward the behavior, subjective norm, and perceived behavioral control [13]. Attitude refers to the degree to which a person has a favorable or unfavorable evaluation of the behavior in question [14]. Subjective norm represents the social pressure an individual perceives from important others (e.g., family, friends, colleagues) to perform or not to perform the behavior [15]. Perceived behavioral control is the individual's perception of their own ability to perform the behavior, which is influenced by factors like resources, opportunities, and skills [16].

TPB has been extensively and fruitfully applied to model and predict a wide range of pro-environmental behaviors. Research across diverse contexts, including waste sorting [17], conservation agriculture [18] and participation in environmental governance [19], consistently demonstrates its robust explanatory power. For instance, a study on rural villagers' willingness to participate in water environment governance in Fujian, China, found that environmental awareness and attitude had a significantly greater impact on participation willingness [20]. Similarly, research on citizens' willingness to pay for urban forest ecosystem services showed that variables such as perceived trustworthiness in the payment mechanism and environmental knowledge, when integrated into the TPB framework, significantly enhanced the model's explanatory power for individuals' willingness to pay [21].

However, a critical limitation of much of the existing TPB literature in the environmental domain is its predominant focus on unidimensional, individual-level analyses. While this approach is effective for understanding isolated behaviors, it often fails to capture the complex dynamics at play within multi-stakeholder collaborative systems, such as those common in ecological governance. This oversight represents a key theoretical gap, as it overlooks how the social context and interpersonal relationships shape an individual's decision to participate.

### The role of SNET in fostering cooperation

While TPB provides a robust model for individual decision-making, SNET perspective—specifically the concept of relational embeddedness—offers crucial insights into the social context that facilitates or hinders cooperation [22,23]. Relational embeddedness refers to the nature and quality of relationships among actors in a network, emphasizing the importance of trust, reciprocity, and shared norms [8]. It is distinct from structural embeddedness, which focuses on an actor's position within a network (e.g., centrality, bridges) [24].

Empirical evidence strongly supports the link between strong, trusting relationships and cooperation. For example, in the healthcare sector, relational embeddedness has been empirically validated as a key antecedent to effective

collaboration and organizational learning [25]. Within ecological governance research, SNET has been productively applied to explicate phenomena related to community-based environmental management and collaborative governance arrangements [26]. For instance, empirical investigations have revealed that the density of trust networks and strength of social connections among community members constitute significant predictors of ecological participation motivation [27].

The core mechanisms through which SNET promotes cooperation involve building trust to reduce transaction costs by mitigating risks and uncertainties associated with cooperation, thereby diminishing reliance on formal contracts and monitoring; facilitating the efficient flow of information, resources and influence among embedded actors, which aligns efforts and fosters collective action; and enforcing cooperative norms through networks founded on trust and reciprocity, which generate social expectations that encourage collaboration and deter free-riding.

In summary, while the TPB explains the formation of individual behavioral intention, the concept of relational embeddedness from SNET elucidates how the social environment supplies the essential mechanisms, grounded in trust, information flow and normative enforcement, for translating such intention into sustained collective and cooperative action, particularly within complex governance contexts such as ecological management.

## Theoretical and empirical rationale for an integrated TPB-SNET framework

The limitations of TPB and SNET when applied in isolation create a compelling rationale for their integration. A review of the literature reveals a growing number of studies that implicitly or explicitly combine social network concepts with behavioral models, providing a foundation for the current study's integrated framework.

Several studies have successfully incorporated social network variables into extended TPB models. For instance, research on farmers' technology adoption has demonstrated that incorporating social network structures and perceived risks into an extended TPB model significantly enhances its explanatory power [28]. Another demonstrated that adding the construct of social network embeddedness to the TPB framework effectively predicts residents' participation intentions [27]. These studies validate the argument that social context variables are not merely external influences but are internalized through the core components of the TPB model.

The integration of the TPB and SNET is well-justified on both theoretical and empirical grounds [29]. The theoretical complementarity is evident, as TPB provides a micro-level mechanism by focusing on individual cognition and intention, while SNET contributes a meso-level perspective by emphasizing the social structures of relational embeddedness that enable and sustain cooperation [28]. Empirical support further strengthens this integration, as studies incorporating social network variables into TPB frameworks have demonstrated improved model fit and explanatory power compared to models based on either theory alone. This approach directly addresses a key research gap, namely, the limitations of unidimensional models in capturing the complexities of multi-stakeholder collaboration. By explicitly modeling how relational embeddedness shapes the core antecedents of behavioral intention, the integrated framework offers a more nuanced and robust explanation of participation dynamics in contexts such as ecological governance.

Table 1 summarizes the core tenets and contributions of the two theories, highlighting the synergistic potential of their integration.

**Table 1. Comparative analysis of the TPB and SNET.**

| Theory | Core concepts | Primary contribution | Limitations |
|--------|---------------|----------------------|-------------|
| TPB | Attitude, Subjective Norm, Perceived Behavioral Control, Intention | Explains individual-level psychological processes leading to behavior [30]. | Fails to account for the influence of social structure and relational context on individual decisions. |
| SNET | Relational Embeddedness, Structural Embeddedness, Trust, Network Ties | Explains how social context and relationships facilitate or constrain cooperation [31]. | Provides less insight into the specific cognitive and motivational processes of individual actors. |

### Research gaps and contributions of this study

Notwithstanding the theoretical utility of both TPB and SNET, a critical knowledge gap persists concerning the specific mechanisms through which relational embeddedness influences pro-environmental behavioral intentions [32]. While it is well-established that relationships matter [33], the precise pathways through which network trust and ties shape attitude, subjective norm and perceived behavioral control remain under-examined, particularly in the context of complex ecological governance.

In response to these theoretical constraints, a significant contribution is made by this study through the development and testing of a novel integrated TPB-SNET analytical framework. This model advances beyond the mere inclusion of network variables within a TPB structure by positing and empirically examining specific mediating and moderating pathways through which relational embeddedness operates. These mechanisms are investigated to yield a more detailed and nuanced understanding of stakeholder participation in ecological governance than has been offered by prior unidimensional or purely structural analyses.

The specific contributions of this research are threefold. First, a rigorous theoretical integration is achieved by bridging individual-level psychological constructs from TPB with the meso-level social structures emphasized in SNET, thereby addressing a notable gap in the literature on collaborative environmental governance. Second, new mechanistic insight is provided through empirical evidence on the specific pathways, such as the mediating effects of attitude and subjective norm. Finally, enhanced predictive power is expected from the integrated framework, as it combines the explanatory strengths of both theories to deliver a more comprehensive and robust explanation of participation behavior within multi-stakeholder ecological governance contexts.

## Hypotheses and theoretical model

### Hypotheses

This section develops hypotheses based on the TPB and SNET, integrating the specific context of ecological co-management at the Lujiangba Coffee Planting Base to systematically deduce the relationships among ENW, SNW, TPB core variables, and Perception of Outcome (PO). Notably, PO is not a substitute for Behavioral Intention (a core component of TPB) but an endogenous variable added in this study to adapt to the ecological co-management context. It refers to stakeholders' subjective predictions and perceptions of ecological, economic and social outcomes resulting from their participation in ecological co-management, aiming to accurately reflect the impact of behavioral value judgments on participation decisions. Its association with TPB core variables aligns with the theoretical logic of "Attitude → Outcome Expectation → Behavioral Intention" [34]. This adaptation of TPB is justified by prior studies: scholars have supplemented TPB with outcome perception variables in ecological behavior research to enhance the model's explanatory power for decision-making mechanisms in multi-stakeholder collaboration scenarios [35].

The ENW is conceptualized as the affective bonds and satisfaction derived from participation in ecological co-management [36], whereas the SNW pertains to the informational and advisory influences that shape decision-making behaviors through social interactions [37].

A critical dimension of relational embeddedness, ENW is posited within SNET to strengthen stakeholders' affective bonds, which in turn may foster an enhanced sense of belonging and responsibility. This strengthened socio-emotional foundation is understood to influence both the attitudes and perceived capabilities of stakeholders engaged in ecological co-management. Simultaneously, SNW, as a principal manifestation of structural embeddedness, operates through the dissemination of information and the reinforcement of social norms, thereby reshaping stakeholders' cognitive frameworks and behavioral dispositions.

Existing studies have mostly examined the impacts of these two types of networks in isolation, lacking a systematic analysis of their synergistic effects and transmission mechanisms through TPB pathways. Thus, the following hypotheses are proposed:

**H1.** Direct impact of ENW and SNW on TPB core variables

**H1-a:** A statistically significant positive relationship is hypothesized between ENW and the IBA of diverse stakeholders. TPB posits that individual attitudes are driven by both emotional experiences and value judgments [38]. From the SNET perspective, ENW reinforces positive perceptions of behavioral value by establishing stable emotional bonds and improving participation satisfaction [39]. Moreover, satisfied emotional needs connect interactive ties to positive behavioral attitudes, buttressing ENW's positive effect on IBA [40]. Such cross-domain findings also validate that emotional network bonds strengthen behavioral value perceptions, underpinning the core logic of H1-a.

**H1-b:** ENW is theorized to exert a positive influence on Subjective Norm (SN). Emotional connections amplify the perceived behavioral expectations of reference groups, reinforcing the effect of ENW on subjective norms [41]. Enhanced emotional connectivity within ENW is anticipated to elevate stakeholders' awareness of prevailing behavioral norms among their peers. Particularly in collaboration among coffee farmers, enterprises and the government, emotional trust facilitates the effective transmission of normative information, making stakeholders more sensitive to the behavioral expectations of other actors.

**H1-c:** ENW has a significant positive impact on the PBC of diverse stakeholders. ENW provides emotional support to stakeholders, which can enhance their confidence in their own participation capabilities. Emotional support from social networks significantly improves farmers' perceived control over ecological planting behaviors [42]. The affective reinforcement provided by ENW is expected to bolster individuals' self-efficacy and perceived capacity to engage in ecological governance initiatives.

**H1-d:** SNW is hypothesized to positively affect IBA. As a carrier of information transmission, SNW optimizes stakeholders' cognitive judgments on ecological co-management through the sharing of professional knowledge and governance experience. Given that TPB posits cognitive optimization as a direct antecedent of individual attitudes [13], informational networks can correct biased perceptions of target behaviors, thereby improving individuals' attitudes toward participation [43,44].

**H1-e:** A significant positive linkage is proposed between the SNW and SN. SNET emphasizes that interactions in structural networks strengthen the binding force of group norms, making individuals more likely to perceive the behavioral expectations of others. Social interactions and information exchange within SNW are posited to amplify individuals' perceptions of normative expectations [44].

**H1-f:** The SNW has a significant positive impact on the PBC of stakeholders. SNW provides abundant information resources and social support, which reduce the costs and risks for stakeholders to participate in ecological co-management, thereby improving their perceived behavioral control [45]. By furnishing informational resources and social support, SNW is theorized to augment stakeholders' confidence in their governance participation capabilities.

**H2:** Direct impact of TPB core variables on behavioral intention

**H2-a:** IBA is hypothesized to exhibit a positive and significant impact on PO. TPB points out that positive individual attitudes drive individuals to form positive predictions of behavioral outcomes [46]. A significant positive correlation exists between positive attitudes toward agricultural ecological behaviors and outcome perceptions [47]. Favorable attitudes toward ecological co-management are anticipated to directly translate into heightened positive perceptions of behavioral outcomes. For example, farmers who recognize ecological co-management are more likely to predict that such behaviors will improve coffee quality and increase income.

**H2-b:** SN has a significant positive impact on PO of stakeholders. Subjective norms reflect individuals' perceptions of others' expectations [48]. When stakeholders perceive that surrounding groups approve of ecological co-management, they tend to predict that the behavior will gain group support and positive outcomes in turn. Stakeholders who perceive strong normative pressure from peers and authorities are more likely to anticipate positive outcomes from participating in ecological co-management.

**H2-c:** PBC is postulated to demonstrate a positive association with PO. Higher PBC means individuals have greater mastery over behaviors, fostering clearer and more positive predictions of behavioral outcomes [49]. For example, farmers

with rich planting experience are more likely to predict that ecological planting will achieve an ecological and economic win-win situation.

**H3:** Direct impact of ENW and SNW on behavioral intention

**H3-a:** The ENW directly influences the PO of stakeholders. Trust relationships reduce information asymmetry, increasing stakeholders' willingness to trust the behavioral outcome experiences shared by others [50]. Affective bonds in ENW facilitate the sharing of authentic experiences of ecological co-management outcomes, enabling stakeholders to form more positive perceptions. For example, trust among coffee farmers in ENW encourages the sharing of real income and ecological benefits from participation, directly shaping others' outcome perceptions.

**H3-b:** The SNW directly influences the PO of stakeholders. Informational exchanges in structural networks provide stakeholders with direct insights into the potential outcomes of ecological co-management, positively shaping their perceptions.

**H4:** Indirect effects of ENW and SNW on behavioral intention.

**H4-a:** ENW is theorized to indirectly shape PO through the mediating roles of IBA, SN, and PBC. ENW first optimizes stakeholders' behavioral attitudes, subjective norm cognition, and perceived behavioral control through emotional bonds, and then further influences their perceptions of behavioral outcomes through the transmission of these three TPB variables.

**H4-b:** SNW is anticipated to exert an indirect influence on PO via the mediation of IBA, SN, and PBC. SNW first reshapes stakeholders' TPB core variables through information transmission and social influence, and then indirectly enhances their positive perceptions of ecological co-management outcomes with the synergistic effect of these variables.

## Theoretical model

Building upon the aforementioned hypotheses, a comprehensive theoretical framework has been constructed to elucidate the underlying mechanisms through which ENW and SNW exert their influence on the participatory behaviors of diverse stakeholders in ecological co-management systems. As demonstrated in Fig 1, the conceptual model integrates multiple dimensions of analysis, thereby offering a systematic representation of the complex interplay among the examined variables.

## Statistical analysis details

**Model specification.** The Partial Least Squares Structural Equation Modeling (PLS-SEM) employed in this study consists of two sub-models: the measurement model (outer model) and the structural model (inner model). Since all

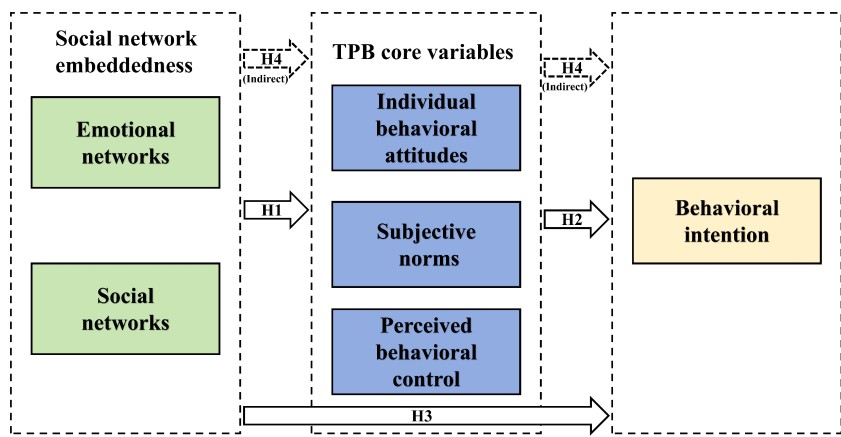

**Fig 1. The conceptual model of this paper.**

latent variables (IBA, SN, PBC, PO, ENW, SNW) are conceptualized as reflective constructs, the Measurement Model is expressed by the linear Equation (1)

$$x_{jh}=\lambda_{jh}\xi_j+\varepsilon_{jh} \tag{1}$$

Where $x_{jh}$ represents the h-th indicator of the j-th latent variable $\xi_j$, $\lambda_{jh}$ is the outer loading (correlation between the indicator and the latent variable), and $\varepsilon_{jh}$ is the measurement error.

The Structural Model, representing the hypothesized relationships in Fig 2, is defined by the following system of linear Equations (2):

$$\xi_{Endogenous}=\sum\beta_{ji}\xi_i+\zeta_j \tag{2}$$

Specifically, for the core outcome variable (PO), the Equation (3) is:

$$PO=\beta_1 IBA+\beta_2 SN+\beta_3 PBC+\beta_4 ENW+\beta_5 SNW+\zeta_{PO} \tag{3}$$

Similarly, the mediating variables are modeled as (equations (4)-(6)):

$$IBA=\gamma_1 ENW+\gamma_2 SNW+\zeta_{IBA} \tag{4}$$

$$SN=\gamma_3 ENW+\gamma_4 SNW+\zeta_{SN} \tag{5}$$

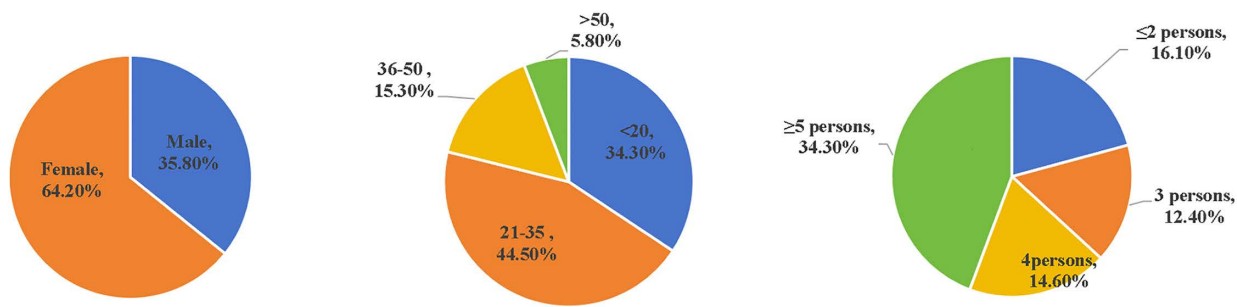

Fig.2-1. Gender distribution of the respondents Fig. 2-2. Age structure of surveyed participants Fig. 2-3. Household size distribution of surveyed participants

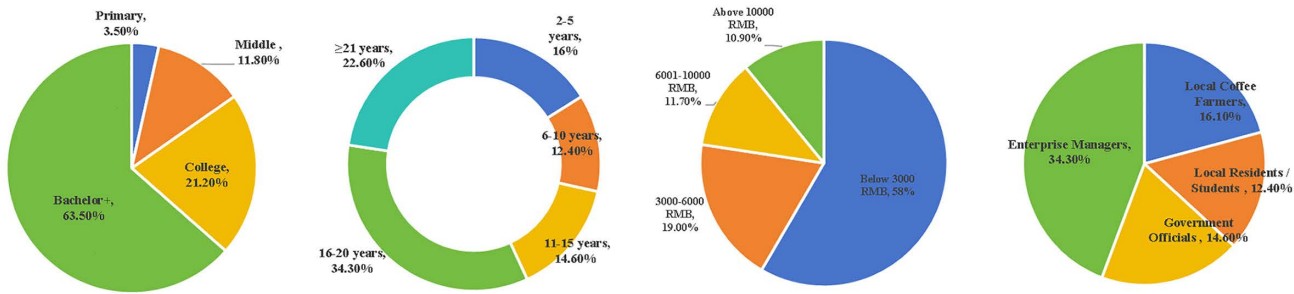

Fig. 2-4. Educational attainment of surveyed participants Fig. 2-5. The number of years the respondents have lived in the study area Fig. 2-6. Personal monthly income of surveyed participants Fig. 2-7. Stakeholder Category

**Fig 2. Demographic results of the survey respondents.**

$$PBC = \gamma_5 ENW + \gamma_6 SNW + \zeta_{PBC} \qquad (6)$$

Where β and γ represent the path coefficients to be estimated, and ζ represents the residual variance (structural error).

**Estimation and objective function.** Unlike covariance-based SEM, PLS-SEM does not optimize a global fit function (such as minimizing the difference between observed and estimated covariance matrices). Instead, it employs an iterative algorithm to maximize the explained variance ($R^2$) of the endogenous latent variables. The objective function is to minimize the residual variances (ζ) in the structural equations.

**Bootstrapping and significance testing.** To assess the statistical significance of path coefficients and indirect effects, this study employed the Bias-Corrected and Accelerated (BCa) Bootstrapping method with 5,000 subsamples under a two-tailed testing framework ($\alpha = 0.05$). The bootstrapping algorithm achieved convergence after three iterations, ensuring the computational stability of the standard error estimates. Specifically, the T-statistics reported in Table 4 calculated as the ratio of the original sample estimate (O) to the standard error (STDEV) derived from the bootstrap distribution (t = O/STDEV), where STDEV represents the standard deviation of the 5,000 bootstrap estimates. Furthermore, the significance of mediation path was assessed by calculating the product of coefficients for each bootstrap subsample. The mean of these products served as the point estimate, while their standard deviation across subsamples functioned as the standard error for the T-test.

**Calculation of latent variable scores.** It is important to clarify that the values of the latent variables used in the structural model were not derived from simple arithmetic means or medians of the observed indicators. Instead, consistent with the PLS-SEM algorithm, latent variable scores were calculated as weighted linear combinations of their respective indicators (Equation (7)):

$$\xi_j = \sum w_{jh} x_{jh} \qquad (7)$$

Where $w_{jh}$ represents the outer weights determined by the PLS algorithm to maximize the explained variance, and $x_{jh}$ represents the standardized values of the observed indicators. This method accounts for the varying contribution of each indicator to the construct, offering higher precision than unweighted averaging.

## Methodology

### Study case

This research examines the Lujiangba Coffee Planting Base in Yunnan Province of China, situated within the Gaoligong Mountains (23°50'-28°30'N, 97°31'-99°05'E), characterized by a subtropical monsoon climate, an annual average temperature of 21.3°C, about 1225 mm of rainfall [51]. The dry-hot valley conditions, particularly at elevations of 780 meters, create an ideal environment for Arabica coffee cultivation [52]. Large-scale coffee estates, including Xinzhai and Biton, have established extensive specialty coffee production bases through the implementation of organic farming practices and modern processing technologies. Concurrently, these enterprises have actively engaged in the promotion of coffee culture. Beyond its agricultural significance, the region is increasingly being recognized as an emerging center for educational tourism. A comprehensive tourism framework has been developed, incorporating guided plantation tours, hands-on coffee processing demonstrations, and structured tasting sessions. These initiatives serve dual purposes: enhancing public awareness of sustainable agricultural practices while simultaneously generating supplementary income streams for local communities. Through the strategic integration of recreational and pedagogical elements, this case demonstrates the potential for agricultural landscapes to simultaneously advance environmental stewardship and socioeconomic development.

The sustainable development observed in this case study is fundamentally supported by rigorous ecological governance mechanisms. Stringent regulatory measures have been implemented to minimize synthetic chemical inputs,

resulting in simultaneous improvements in both coffee quality and ecosystem integrity. This ecological emphasis has been effectively leveraged to enrich the tourism sector, with educational programming explicitly linking environmental conservation principles to practical agricultural applications.

## Data

**Questionnaire design.** The questionnaire was designed based on the TPB and the concept of social network embeddedness. It aimed to explore the influence of SNW and ENW on the ecological co-management behavior of diverse stakeholders in the Lujiangba coffee plantation base. The questionnaire consisted of six latent variables: individual behavior attitude (IBA), subjective norm (SN), perceived behavioral control (PBC), perception of the behavioral outcome (PO), emotional network (ENW), and suggestive networks (SNW).

Before the formal investigation, a pilot test was conducted on a small scale. The purpose of the pilot test was to assess the feasibility and appropriateness of the questionnaire. The questionnaire was distributed to stakeholders involved in ecological governance in the Lujiangba coffee plantation base, including local residents, government officials, environmental organizations and coffee plantation managers. It aimed to identify any issues with the wording, structure, or clarity of the questions, as well as to estimate the time required for respondents to complete the survey. This process helped to refine the questionnaire and ensure that it was comprehensible and relevant to the target respondents.

A total of 85 questionnaires were distributed for this preliminary survey. Through the reliability and validity analysis of the questionnaires, it was found that the overall reliability and structural validity of each scale reached an acceptable level, confirming the feasibility for further research. The results of the pilot test of the questionnaire are detailed in S1 Appendix (Tables A1-A2). In terms of reliability, the Cronbach's α coefficients of all latent variables exceeded the threshold of 0.7, with IBA (0.912), SN (0.883), and PBC (0.970) demonstrating high internal consistency, indicating that the observed variables can reliably reflect the characteristics of the latent variables. The reliability coefficient for ENW was 0.795, which meets basic requirements but is slightly lower compared to other dimensions. Therefore, prior to the formal survey, the study reviewed the phrasing of its observed variables (ENW1-ENW3) and made revisions to the item wording, further improving the questionnaire. The validity analysis showed significant results for the Bartlett's test of sphericity ($\chi^2 = 1338.868$, $p < 0.01$), indicating that the data is suitable for factor analysis. The rotated component matrix further confirmed the structural validity of the scales, with the factor loadings of all observed variables on their corresponding components exceeding 0.7 (e.g., IBA1=0.803, SN1=0.811), and the communalities generally exceeding 0.6, suggesting that the variables can effectively explain the variance of the latent variables. The adjusted questionnaire of this study is shown in S1 Appendix (Table A3).

All measurement items were assessed using a 5-point Likert scale, where 1 represented 'Strongly Disagree' and 5 represented 'Strongly Agree'. The data were recorded exactly as selected by the respondents without any transformation, truncation, or modification to the original values. regarding data completeness, the final dataset contained no missing values, as incomplete questionnaires were excluded during the data cleaning phase prior to analysis.

**Data acquisition and sampling strategy.** The formal investigation was conducted from July 2024 to March 2025 using a stratified purposive sampling method to ensure the representation of multiple stakeholders involved in the ecological co-management of the Lujiangba coffee plantation base.

To ensure the diversity of the sample, the target population was divided into four distinct stakeholder groups: (1) Local coffee farmers, who are the direct practitioners of ecological cultivation; (2) Enterprise managers, representing coffee processing factories and estates; (3) Government officials, including staff from the local agricultural bureau and environmental protection station; and (4) Local residents, who are affected by the ecological environment.

To minimize sampling bias, the recruitment procedure employed a mixed-method approach specifically tailored to the characteristics of each stakeholder group. For farmers and residents, face-to-face interviews were conducted in the fields and community centers of Xinzhai and Biton villages, a strategy designed to overcome potential digital barriers. In parallel, the research team visited major coffee estates, such as Xinzhai and Biton, to distribute questionnaires to enterprise

managers, while surveys for government officials were administered through local administrative offices. Furthermore, an electronic version of the survey via Wenjuanxing was utilized as an online supplement to effectively reach younger stake-holders and tourists engaged in the coffee culture experience.

A total of 146 questionnaires were distributed. After excluding invalid responses (e.g., incomplete answers or uniform selection), 137 valid questionnaires were obtained, resulting in an effective response rate of 93.8%.

This study was conducted in accordance with the Declaration of Helsinki and was reviewed and approved by the Ethics Committee of Baoshan University. Informed consent was obtained from all individual participants included in the study. For the face-to-face interviews, verbal informed consent was recorded prior to the survey to ensure anonymity. For the online survey, a consent statement was presented on the first page, and participants indicated their consent by clicking "Agree" to proceed with the questionnaire. All data were anonymized and used solely for academic research purposes.

**Descriptive analysis of demographic characteristics.** A comprehensive descriptive statistical analysis was conducted based on survey data obtained from 137 respondents through an online questionnaire. As illustrated in Fig 2-1, the gender distribution reveals that 35.8% of the participants were male, while females constituted a predominant proportion of 64.2%. With respect to age distribution (Fig 2-2), although the 21–35 age group is presumed to represent the largest cohort, precise data are unavailable due to missing values; among the valid responses, 34.3% were under 20 years of age, 15.3% belonged to the 36–50 age group, and 5.8% were over 50 years old.

Household composition (Fig 2-3) was characterized as follows: 5.1% of respondents resided in households comprising two or fewer individuals, 14.6% in three-person households, 37.2% in four-person households, and 43.1% in households

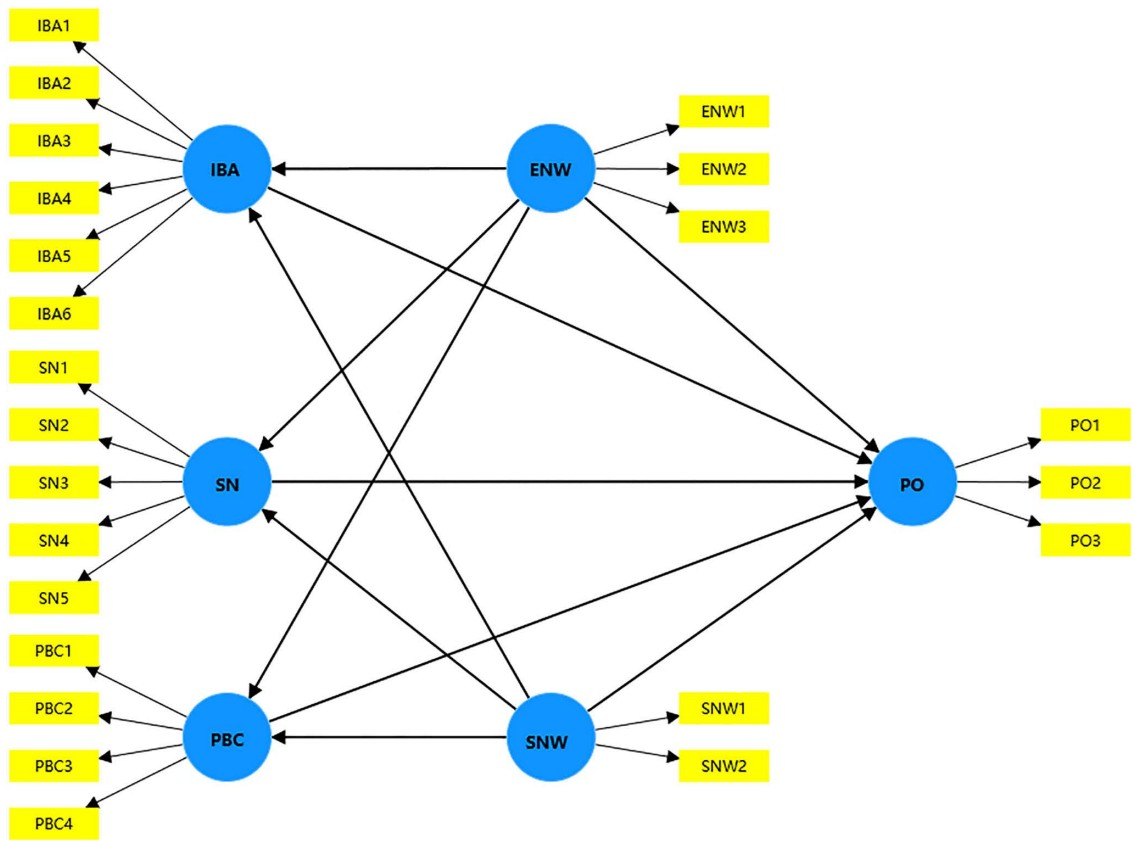

**Fig 3. Paths of structural equation model.**

with five or more members. Concerning educational attainment (Fig 2-4), the majority (63.5%) possessed a bachelor's degree or higher, followed by 21.2% with a college diploma, 11.8% who had completed middle school, and 3.5% with only primary education.

The duration of residence within the study case (Fig 2-5) exhibited considerable variation: 16% of participants had lived in the study case for 2–5 years, 12.4% for 6–10 years, 14.6% for 11–15 years, 34.3% for 16–20 years, and 22.6% for 21 years or longer. Monthly income distribution (Fig 2-6) further demonstrated that 58.4% of respondents earned less than 3,000 RMB, 19.0% received between 3,000–6,000 RMB, 11.7% fell within the 6,001–10,000 RMB range, and 10.9% reported incomes exceeding 10,000 RMB. Crucially, regarding the distribution of stakeholder categories, the sample effectively covers the diverse actors required for the co-management analysis (Fig 2-7). Based on the socioeconomic characteristics of the respondents, the sample comprised 63 local coffee farmers (46.0%), who constitute the core body of ecological practice; 15 enterprise managers (10.9%) from coffee estates and processing factories; 16 government offi-cials (11.7%) involved in local administration and environmental regulation; and 43 local residents and students (31.4%), representing the broader community and educational tourism participants. This balanced distribution validates the study's premise of 'multi-subject' participation and ensures that the perspectives of regulators, producers, and beneficiaries are all adequately represented. Collectively, these statistics provide a detailed profile of the demographic, educational, and socio-economic characteristics of the study participants. The demographic profile of survey respondents yields critical insights for fostering multi-stakeholder engagement in ecological co-management at the Lujiangba coffee plantation base. The pronounced female predominance in the sample underscores the pivotal role of women in local ecological management practices. The prevalence of younger individuals within the age distribution signals the imperative of incorporating youth perspectives to ensure the long-term viability of sustainability initiatives. The relatively high educational attainment among participants suggests a capacity for the application of advanced ecological knowledge in governance frameworks. Varia-tions in household size and income levels further highlight the necessity of accounting for socioeconomic diversity when formulating governance strategies.

## Results and discussions

### Measurement model

The measurement properties of the proposed model were rigorously examined through a two-step structural equa-tion modeling (SEM) approach, utilizing SmartPLS 4.0 statistical software for analysis. Convergence was achieved during the bootstrapping procedure upon completion of the fifth iteration. With respect to reliability assessment, it was observed that all latent constructs demonstrated satisfactory internal consistency, as evidenced by Cronbach's α coef-ficients surpassing the threshold of 0.75. Furthermore, composite reliability (CR) values were found to range between 0.884 and 0.934, thereby exceeding the recommended benchmark of 0.8 and confirming the robustness of the mea-surement model. Convergent validity was established through multiple indicators. All observed variables exhibited fac-tor loadings exceeding 0.7 (range: 0.790–0.910), while the average variance extracted (AVE) values for each construct surpassed the 0.5 criterion (range: 0.693–0.943). These findings collectively support the presence of adequate conver-gent validity, as presented in Table 2.

Regarding discriminant validity, the Fornell-Larcker Criterion was employed for verification. The analysis revealed that the square roots of AVE values for all latent factors consistently exceeded their corresponding correlation coefficients with other constructs (Table 2), thereby confirming that discriminant validity requirements were satisfactorily met (Table 3).

### Structural model evaluation

In the assessment of the structural model within this investigation, potential multicollinearity among latent constructs was examined through the utilization of the Variance Inflation Factor (VIF) derived from PLS-SEM, which served as the primary

**Table 2. Results of the measurement model.**

| Latent Variables | Indicators | Factor Loadings | AVE | CR | Cronbach's α |
|---|---|---|---|---|---|
| IBA | IBA1 | 0.826 | 0.701 | 0.934 | 0.915 |
| | IBA2 | 0.862 | | | |
| | IBA3 | 0.834 | | | |
| | IBA4 | 0.811 | | | |
| | IBA5 | 0.859 | | | |
| | IBA6 | 0.830 | | | |
| SN | SN1 | 0.853 | 0.693 | 0.919 | 0.890 |
| | SN2 | 0.860 | | | |
| | SN3 | 0.815 | | | |
| | SN4 | 0.841 | | | |
| | SN5 | 0.792 | | | |
| PBC | PBC1 | 0.865 | 0.697 | 0.902 | 0.855 |
| | PBC2 | 0.844 | | | |
| | PBC3 | 0.771 | | | |
| | PBC4 | 0.855 | | | |
| PO | PO1 | 0.975 | 0.943 | 0.980 | 0.970 |
| | PO2 | 0.974 | | | |
| | PO3 | 0.964 | | | |
| ENW | ENW1 | 0.883 | 0.718 | 0.884 | 0.804 |
| | ENW2 | 0.850 | | | |
| | ENW3 | 0.808 | | | |
| SNW | SNW1 | 0.929 | 0.855 | 0.922 | 0.830 |
| | SNW2 | 0.920 | | | |

**Table 3. Discriminant validity results (Fornell-Larcker Criterion).**

| | ENW | SNW | IBA | SN | PBC | PO |
|---|---|---|---|---|---|---|
| ENW | **0.848** | | | | | |
| SNW | 0.209 | **0.924** | | | | |
| IBA | 0.311 | 0.297 | **0.837** | | | |
| SN | 0.301 | 0.246 | 0.177 | **0.833** | | |
| PBC | 0.287 | 0.33 | 0.213 | 0.129 | **0.834** | |
| PO | 0.451 | 0.465 | 0.471 | 0.385 | 0.447 | **0.971** |

Note: Bold values represent the square root of AVE, and others are correlation coefficients.

diagnostic criterion. PLS-SEM has been widely recognized for its distinct advantages in the analysis of survey data characterized by limited sample sizes, owing to its capacity to mitigate sample size dependencies via iterative computational algorithms [53]. Furthermore, this methodological approach demonstrates robustness in accommodating complex model configurations, including those involving multiple latent variables and mediating effects, while simultaneously obviating the necessity for stringent distributional assumptions [54].

The diagnostic outcomes revealed that the VIF values associated with indicators corresponding to all latent constructs fell within a range of 1.595 to 9.263. Notably, the VIF metrics for ENW, IBA, SN and SNW were observed to be substantially lower than the conventional threshold of 10. Although the VIF value pertaining to PO exhibited a comparatively elevated magnitude, it nevertheless remained below the empirically established cutoff. Consequently, the absence of

pronounced multicollinearity concerns among the latent constructs within the structural model was confirmed, thereby satisfying the statistical prerequisites for ensuing path analytical procedures.

To evaluate the statistical significance of path relationships within the structural equation model, the BCa Bootstrap method was implemented, employing a two-tailed testing framework with a predetermined significance level of 0.05. The analytical process achieved convergence following three iterations, as substantiated by the outputs presented in Fig 3 and Table 4.

### Direct effects

The direct influence pathways among the indicators, as illustrated in Fig 3, are systematically presented in Table 4. Based on the empirical evidence, the following conclusions can be substantiated.

### Hypothesis H1 validation

The structural equation modeling results reveal a statistically significant positive relationship between ENW and IBA ((t = 3.27, p = 0.001)). This finding is consistent with the theoretical proposition that emotional networks facilitate the formation of positive attitudes toward ecological co-management.This finding is consistent with the theoretical propositions of SNWT, which posits that emotional fulfillment and enhanced community belonging derived from network participation contribute to the development of favorable cognitive evaluations regarding co-management behaviors. Consequently, Hypothesis H1-a receives robust empirical support.

Furthermore, the analysis indicates that ENW is significantly and positively associated with SN (t = 3.196, p = 0.001). This discrepancy can be theoretically explicated by social identity theory and environmental psychology research: ENW foster in-group cohesion and value alignment through affective bonds (e.g., trust, satisfaction, and shared experiences), which amplify individuals' sensitivity to normative expectations within the community [55]. In ecological co-management contexts, ENW-derived emotional attachments strengthen stakeholders' identification with the collective environmental goals, making them more likely to internalize group norms as personal standards [56] For instance, participation in community-led tree-planting events or environmental clean-ups nurtures a sense of "we-ness," which reinforces the perceived social pressure to engage in pro-ecological behaviors, thus explaining ENW's stronger impact on SN.

This relationship suggests that emotional networks amplify the impact of social normative pressure by reinforcing individuals' perceptions of prevailing group norms. For instance, participaton in community collective activities within

**Table 4. Path coefficients of direct influence and corresponding significance levels.**

| Path relationships | Coefficients | t-value | p-value |
| --- | --- | --- | --- |
| ENW -> IBA | 0.261** | 3.27 | 0.001 |
| ENW -> PBC | 0.229** | 2.854 | 0.004 |
| ENW -> PO | 0.198** | 2.89 | 0.004 |
| ENW -> SN | 0.261** | 3.196 | 0.001 |
| IBA -> PO | 0.259*** | 3.654 | 0 |
| PBC -> PO | 0.237** | 3.169 | 0.002 |
| SN -> PO | 0.195** | 2.737 | 0.006 |
| SNW -> IBA | 0.242** | 3.132 | 0.002 |
| SNW -> PBC | 0.282*** | 3.66 | 0 |
| SNW -> PO | 0.22** | 3.275 | 0.001 |
| SNW -> SN | 0.192* | 2.464 | 0.014 |

Note: *** denotes p < 0.001, ** denotes p < 0.01, and * denotes p < 0.05.

emotional networks has been shown to heighten awareness of peer support for ecological initiatives, thereby intensifying the effect of subjective norms. These results provide conclusive evidence supporting Hypothesis H1-b.

Regarding perceived behavioral control, the path coefficient from ENW to PBC (t = 2.854, p = 0.004) reaches statistical significance. This outcome substantiates the mechanism whereby emotional connections within networks enhance participants' self-efficacy beliefs through strengthened community identity and responsibility. The empirical data, including responses to the ENW3 measurement item, confirm that positive affective experiences reduce psychological barriers to participation. These findings align with the theoretical framework of social network embeddedness, thereby validating Hypothesis H1-c.

The analytical results further demonstrate that SNW shows a significant positive correlation with IBA (t = 3.132, p = 0.002). Interpreting this through the lens of SNET, we suggest this relationship may be driven by the dual mechanisms of information dissemination and social influence operating within suggestive networks, where inputs from community opinion leaders substantially shape participants' cognitive frameworks. These results corroborate the fundamental premise of the Theory of Planned Behavior (TPB) regarding attitude formation, thus confirming Hypothesis H1-d.

The path coefficient from the SNW to SN is 0.192 (t = 2.464, p = 0.014), significant at the 0.05 level. This finding implies that information exchange within social networks contributes to the internalization of group norms, as exemplified by the consensus-building function of community bulletin boards. Despite the coefficient's magnitude being comparatively lower than that of emotional networks, its significance threshold provides sufficient evidence to accept Hypothesis H1-e, in accordance with TPB's normative pressure postulates.

The path coefficient from the SNW to PBC is 0.282 (t = 3.66, p < 0.001), significant at the 0.001 level. This robust association is theoretically grounded in the information-processing perspective of environmental psychology and social network theory: SNW specialize in transmitting practical knowledge, technical guidance and resource-related information that directly enhances stakeholders' perceived capability to engage in co-management [57]. Unlike ENW, which operates through affective empowerment, SNW reduces perceived behavioral barriers by addressing informational asymmetries and building functional competence [58]. For example, SNW indicators capturing the dissemination of sustainable coffee cultivation techniques or government subsidy policies equip stakeholders with tangible resources and skills, thereby strengthening their confidence in executing ecological management tasks [59]. This robust association indicates that informational networks substantially enhance participants' perceived competence through practical knowledge transfer and support systems, including skill development programs and policy clarification initiatives. The effectiveness of case study dissemination (SNW2 item) in overcoming participation barriers provides empirical validation for the information embeddedness mechanism, thereby conclusively supporting Hypothesis H1-f.

### Hypothesis H2 verification

The analytical outcomes reveal that IBA serves as the strongest correlate of PO (t = 3.654, p < 0.001) within the TPB framework, a finding that strongly corroborates the central tenet of TPB regarding attitude-behavior consistency. The measurement robustness of the IBA construct (AVE = 0.701) further reinforces the validity of Hypothesis H2-a.

While demonstrating a comparatively modest effect size, the SN-PO relationship (t = 2.737, p = 0.006) nevertheless achieves statistical significance. This result confirms the supplementary role of normative pressure in intention formation, consistent with previous research emphasizing the motivational influence of group identification. Thus, Hypothesis H2-b receives empirical confirmation.

The analysis identifies PBC as another significant antecedent of PO (t = 3.169, p = 0.002), indicating that resource availability and capability perceptions substantially influence participation intentions. Beyond verifying TPB's core proposition, this finding elucidates the indirect pathways through which network embeddedness operates, thereby providing comprehensive support for Hypothesis H2-c.

## Hypothesis H3 confirmation

The structural model yields significant direct effects for both network types on participation intention, thereby validating Hypothesis H3. Specifically, the ENW-PO path (t=2.890, p=0.004) confirms that affective network ties directly motivate engagement, independent of cognitive or control mediators.

This direct effect reflects the "affective priming" mechanism in environmental psychology: emotional bonds derived from ENW can directly trigger pro-environmental intentions without requiring cognitive mediation, as affect often precedes deliberate decision-making [60].

This relationship is particularly evident in the ENW1 measurement dimension, which captures the participation-enhancing effect of emotional bonds, thus substantiating H3-a.

Similarly, the SNW-PO relationship (t=3.275, p=0.001) demonstrates that informational networks directly facilitate intention formation through goal alignment mechanisms. This empirical evidence provides conclusive support for H3-b.

## Indirect effects: Hypothesis H4 examination

In addition to the direct relationships previously examined, the mediation pathways among the indicators depicted in Fig 3 have been systematically analyzed, with the comprehensive results presented in Table 5.

The mediation analysis reveals a statistically significant indirect effect of ENW on PO through IBA (t=2.341, p=0.019). This finding suggests that positive attitudes toward ecological co-management are cultivated through emotional connections within the network, thereby serving as a psychological mechanism that enhances participation willingness. Specifically, the affective components captured by the ENW2 measurement item appear to coincide with the formation of favorable cognitive evaluations regarding co-management behaviors. These results align with the theoretical propositions of SNET, which posits that emotional networks amplify attitude-behavior consistency through the reinforcement of community belongingness.

Furthermore, the ENW→SN→PO pathway (t=2.052, p=0.04) has been identified as another significant mediation route. This relationship implies that participation in collective activities within emotional networks, such as community-based environmental initiatives, serves to strengthen group identity and normative perceptions. Consequently, individuals become more sensitive to prevailing social expectations, which in turn increases their behavioral intention. These observations provide empirical support for the theoretical mechanism whereby emotional embeddedness facilitates norm internalization, as postulated by TPB.

With respect to perceived behavioral control, the ENW→PBC→PO pathway (t=2.071, p=0.038) has been confirmed as statistically significant. This result indicates that the psychological barriers to participation are effectively mitigated through positive emotional experiences within the network, as evidenced by responses to the PBC1 measurement item. Such findings corroborate the SNET framework's assertion regarding the empowerment function of emotional connections in enhancing individuals' perceived competence and resource availability.

The analysis of suggestive network effects yields a significant indirect relationship through the SNW→IBA→PO pathway (t=2.088, p=0.037). This outcome can be attributed to the cognitive restructuring that occurs through exposure to

Table 5. Statistical results of mediation effects analysis.

| Mediation pathways | Sample mean (M) | T Statistics (|O/STDEV|) | p-value |
| --- | --- | --- | --- |
| SNW→PBC→PO | 0.067 | 2.341 | 0.019 |
| ENW→IBA→PO | 0.067 | 2.399 | 0.016 |
| ENW→SN→PO | 0.051 | 2.052 | 0.040 |
| SNW→IBA→PO | 0.063 | 2.088 | 0.037 |
| SNW→SN→PO | 0.037 | 1.731 | 0.084 |
| ENW→PBC→PO | 0.054 | 2.071 | 0.038 |

informational content, such as that measured by the SNW1 item. According to TPB's theoretical framework, such information-dissemination serves to reshape attitudes through social influence processes, thereby establishing an information-mediated pathway to behavioral intention formation. Practical manifestations of this mechanism include the enhanced recognition of governance value through exposure to community bulletin board content.

The indirect effect path coefficient from the SNW to PO via SN is 0.037 (t = 1.731, p = 0.084), which does not reach the 0.05 significance level. This suggests that while the suggestive network partially enhances perception of group norms (e.g., the SN4 item), its indirect impact on behavioral intention is weak, possibly due to variations in social identity among information recipients. For instance, low-identity groups may respond less to normative information from the SNW.

Notably, the most robust mediation effect is observed in the SNW→PBC→PO pathway, supported by highly significant constituent paths (SNW→PBC: p < 0.001; PBC→PO: p = 0.002). This finding underscores the critical role of practical support mechanisms (exemplified by the SNW2 item) in building participants' self-efficacy beliefs. These results validate the mechanism of information embeddedness enhances self-efficacy in social networks.

To further elucidate the distinction between direct and indirect pathways, a theoretical framework grounded in dual-process models of decision-making is proposed [61]: This framework delineates two principal mechanisms: (1) Direct effects (ENW→PO and SNW→PO) are characterized by automatic, affect- or information-driven processes that circumvent deliberate cognitive evaluation. In this pathway, pre-existing emotional bonds are leveraged by ENW to elicit immediate pro-intentions, whereas goal-relevant information is utilized by SNW to align intentions without mediation through attitudinal or control-related constructs. (2) Indirect effects (e.g., ENW→IBA→PO, SNW→PBC→PO), in contrast, entail deliberative cognitive processing wherein core constructs of the attitudes, subjective norms and perceived behavioral control, are initially shaped by network influences prior to their impact on behavioral intentions. For instance, indirect influence of ENW on PO is mediated through the fostering of positive attitudes, which reflect a cognitive appraisal of co-management benefits. Similarly, SNW operates indirectly by enhancing perceived behavioral control, defined as a cognitive assessment of one's capability to perform the behavior [62]. This dual-process distinction is consistent with prior environmental psychology research, which posits that pro-environmental intentions may emerge from both automatic (implicit) and deliberate (explicit) psychological mechanisms [63]. Consequently, the present framework advances a more nuanced understanding of the complementary pathways through which social networks shape participatory behavior in ecological co-management contexts.

## Sensitivity analysis of statistical significance

To address the potential issue of alpha inflation arising from multiple hypothesis testing (comprising 11 direct and 6 indirect paths), a post-hoc sensitivity analysis was conducted using the Bonferroni correction. With an overall significance level set at 0.05, the adjusted threshold for individual path significance was established at approximately 0.0029 (0.05/17). The results, as summarized in Table 4, indicate that seven core direct pathways demonstrate high robustness under this conservative criterion: IBA→PO (p < 0.001), SNW→PBC (p < 0.001), ENW→SN (p = 0.001), ENW→IBA (p = 0.001), SNW→PO (p = 0.001), SNW→IBA (p = 0.002), and PBC→PO (p = 0.002). These relationships represent the fundamental structure of our theoretical model and remain statistically significant even under the most stringent error control.

Although four additional direct paths (ENW→PO, ENW→PBC, SN→PO, SNW→SN) and five indirect mediation paths (Table 5) yielded p-values within the range of 0.003 to 0.05, they are retained as meaningful findings in this study. This decision is supported by the fact that the Bonferroni correction is often considered overly conservative for structural equation modeling (SEM), where paths are theoretically interdependent rather than isolated. Furthermore, given the exploratory nature of applying SNET to ecological co-management, prioritizing the avoidance of Type II errors is essential to capture emerging social mechanisms. Consequently, we interpret these results with transparency: while the core structural links are exceptionally robust, the secondary effects and specific mediation mechanisms should be viewed as suggestive evidence within the standard 95% confidence interval, interpreted within the broader context of experiment-wise error rates.

## Discussions

This investigation extends the theoretical boundaries of the TPB and SNET through their novel application in ecological co-management research. By incorporating both ENW and SNW into the analytical framework, a more comprehensive understanding of social network mechanisms has been achieved. The empirical findings reveal a sophisticated dual-pathway mechanism through which network embeddedness operates: direct influences on participation intentions are complemented by indirect effects mediated through IBA, SN and PBC.

The analysis indicates that both types of social networks have significant positive associations with the key TPB constructs. These findings corroborate existing literature in environmental psychology that emphasizes the formative role of social networks in shaping environmental attitudes and behaviors [64]. Particularly noteworthy is the robust influence of SNW on both IBA and PBC, which substantiates the theoretical proposition that informational networks serve as critical channels for knowledge dissemination and capability building in environmental governance contexts Similarly, the significant effect of SNW on IBA and PBC supports the notion that information and social influence mechanisms play a crucial role in enhancing stakeholder participation [65]. Furthermore, Consistent with the foundational principles of TPB, all three mediating variables were found to be significantly associated with participation intentions [66]. The direct effects of ENW and SNW on PO further emphasize the importance of social networks in driving stakeholder participation intentions. This finding is consistent with studies that highlight the role of social networks in promoting collective action and environmental stewardship [67]. The study identifies significant indirect effects of ENW and SNW on PO through IBA, SN, and PBC. These findings align with research that highlights the mediating roles of IBA, SN, and PBC in the relationship between social networks and behavioral intentions [68].

The theoretical contributions of this study are threefold. First, a clarification is provided regarding the differential mechanisms through which ENW and SNW shape the constructs of the TPB. Specifically, ENW is found to exert a stronger influence on subjective norms via social identity and affective bonding, whereas SNW is more closely associated with perceived behavioral control through informational support and competence development. This distinction addresses a gap in the extant literature, wherein social networks have frequently been conceptualized as a unidimensional construct. Second, by integrating dual-process models of decision-making with TPB and SNET, a theoretical explanation is advanced for the distinct direct and indirect pathways through which network embeddedness influences participatory intentions. This integrated framework yields a more nuanced understanding of social influence mechanisms within the context of ecological governance. Third, the empirical findings validate the cross-context applicability of social identity theory and self-efficacy theory within agricultural-ecological co-management systems, thereby extending their relevance beyond conventional environmental psychology research domains.

In conclusion, this study contributes to the literature by furnishing empirical evidence on the multidimensional role of social network embeddedness in ecological co-management and by offering a theoretically grounded explanation for the differential effects exerted by emotional versus suggestive networks. The findings carry practical implications for the design of targeted interventions aimed at enhancing stakeholder participation. It is recommended that policymakers and practitioners prioritize the cultivation of emotional bonds to reinforce normative compliance. Concurrently, investments in informational networks, such as training programs and structured policy dissemination, are advised to strengthen stakeholders' perceived behavioral control.

## Conclusions and policy recommendations

### Conclusions

This study advances the integration of the TPB and SNET by empirically elucidating the multidimensional mechanisms through which social networks shape multi-stakeholder ecological co-management intentions within agri-tourism communities. Building upon the TPB's focus on cognitive antecedents, i.e., attitudes, subjective norms, and perceived behavioral

control and SNET's emphasis on relational embeddedness, it is demonstrated that emotional networks and suggestive networks operate through distinct yet complementary pathways to influence participation intentions. This theoretical extension addresses the historical separation between cognitive and relational perspectives in environmental behavior research.

Specifically, the core proposition of the TPB is validated, confirming that individual behavioral attitudes, subjective norms, and perceived behavioral control serve as key predictors of behavioral intentions, with individual behavioral attitudes emerging as the strongest driver. However, through the integration of SNET, it is further revealed that social networks amplify the explanatory power of the TPB via dual-process mechanisms. First, automatic pathways are identified, wherein emotional networks trigger affective priming and suggestive networks provide cognitive clarity, thereby bypassing deliberate cognitive mediation. Second, deliberative pathways are observed, whereby emotional networks strengthen subjective norms through social identity formation, while suggestive networks enhance perceived behavioral control via informational support.

These findings refine SNET by distinguishing between emotional embeddedness and informational embeddedness. It is shown that emotional networks exert a stronger influence on subjective norms, whereas suggestive networks are more closely linked to perceived behavioral control, a nuanced distinction that extends the understanding of how different network functions shape the constructs of the TPB.

From a practical perspective, the results highlight that intrinsic motivation and self-efficacy outweigh normative pressure in driving ecological co-management intentions within agricultural settings. This underscores the need to prioritize attitude change and capability building alongside interventions based on social norms. By bridging the TPB and SNET, a unifying framework is provided for explaining multi-stakeholder environmental behavior, demonstrating that social networks do not merely supplement cognitive factors but actively reshape them through affective and informational mechanisms.

This study investigated the behavioral driving mechanisms of multi-stakeholder ecological co-management in a China's coffee-producing area by integrating the TPB and SNET. The results demonstrate that both ENW and SNW significantly enhance stakeholders' participation intentions through direct and indirect pathways. Specifically, higher levels of ENW are associated with stronger IBA, SN, and PBC by fostering emotional bonds and collective identity. Meanwhile, SNW is positively linked to IBA, SN, and PBC, potentially through pathways of information dissemination. It is worth noting that, descriptively, both IBA and PBC exhibited larger path coefficients regarding participation intentions compared to SN, highlighting the importance of intrinsic motivation and self-efficacy in ecological co-management. This study advances the TPB by incorporating SNET, revealing how social networks shape behavioral intentions beyond traditional cognitive factors. It bridges a critical gap in multi-stakeholder governance research by systematically analyzing relational embeddedness rather than focusing solely on structural network properties.

## Policy recommendations

Guided by empirical findings on the differential mechanisms of emotional networks and suggestive networks, along with their targeted effects on the constructs of the Theory of Planned Behavior, evidence-based and context-specific policy recommendations are proposed. These recommendations are tailored to the ecological co-management needs of agricultural communities, moving beyond broad governance frameworks toward actionable interventions aligned with both theoretical and statistical insights.

To strengthen normative internalization and emotional commitment, the reinforcement of emotional networks is recommended. Given the stronger influence of emotional networks on subjective norms through social identity formation, policy interventions should prioritize the building of affective bonds to enhance normative compliance. Community-based emotional bonding programs can be designed, featuring regular ecology-themed collective activities, such as joint forest restoration and communal composting projects, which integrate environmental stewardship with social interaction. Additionally, affective recognition systems, such as village-level ecological stewardship champions programs, can be implemented to

publicly honor active participants through ceremonies or local media. Shared narrative building through local storytelling campaigns, including oral histories of environmental conservation and community-led documentaries, is also encouraged to cultivate emotional commitment to communal ecological goals.

To enhance individual behavioral attitudes and perceived behavioral control, the optimization of suggestive networks is advised. Leveraging the robust association of suggestive networks with perceived behavioral control and its significant effects on individual behavioral attitudes, policies should focus on informational support and capacity building. Multi-channel, practical information dissemination can be established through an ecological co-management information hub that integrates digital platforms and traditional channels to deliver targeted content. This includes data on local ecological benefits to reshape attitudes toward co-management, as well as step-by-step guides on sustainable cultivation techniques and subsidy application processes to address informational asymmetries. Furthermore, opinion leader engagement can be promoted by training local coffee farmers, enterprise managers, and village leaders as ecological ambassadors to disseminate information and model pro-environmental behavior. Practical skill-building workshops, offering hands-on training on ecological management tailored to agricultural stakeholders' needs, are also recommended to directly enhance perceived behavioral control.

Finally, the establishment of a network-centered co-management mechanism is proposed to integrate the effects of emotional and suggestive networks into governance structures. Stakeholder network mapping should be conducted to systematically identify existing emotional and suggestive networks, key actors, and areas where networks are weakest. Participatory decision-making forums can be created to hold regular, inclusive meetings that combine social interaction with information exchange, ensuring that both emotional bonds and practical knowledge inform policy design. Moreover, cross-sectoral network coordination should be mandated to align community activities focused on emotional networks with capacity-building efforts focused on suggestive networks, thereby creating a synergistic system that addresses both affective and cognitive barriers to participation.

## Limitations

Several limitations of this study should be acknowledged. First, the generalizability of the findings may be constrained by the relatively small sample drawn from a single coffee-producing region, which limits the applicability of the results to agricultural communities with distinct socioeconomic or geographical characteristics. Second, the measurement of emotional and suggestive networks relies on self-reported data, which could be influenced by social desirability bias, and the cross-sectional design precludes the establishment of causal relationships between network mechanisms and behavioral intentions. Third, while relational embeddedness is examined, structural properties of the networks are not accounted for, nor are potential moderating effects of individual differences across stakeholder types. Addressing these limitations in future research would strengthen the theoretical and practical contributions of this line of inquiry.

## Nomenclature

| TPB | Theory of planned behavior |
|-----|----------------------------|
| SNET | Social network embeddedness theory |
| SEM | Structural equation modeling |
| IBA | Individual behavioral attitudes |
| SN | Subjective norms |
| PBC | Perceived behavioral control |
| PO | Perception of the behavioral outcome |
| ENW | Emotional networks |
| SNW | Suggestive networks |
| CR | Composite reliability |

| TPB | Theory of planned behavior |
|-----|----------------------------|
| AVE | Average variance extracted |
| VIF | Variance inflation factor |
| PLS-SEM | Partial Least Squares Structural Equation Modeling |

## Supporting information

**S1 Appendix. Appendix tables.**
(DOCX)

## Author contributions

**Conceptualization:** Xiumei Xu.

**Data curation:** Xiumei Xu, Dengke Wang.

**Formal analysis:** Xiumei Xu, Dengke Wang.

**Funding acquisition:** Xiumei Xu.

**Investigation:** Xiumei Xu.

**Methodology:** Xiumei Xu, Dengke Wang.

**Software:** Dengke Wang.

**Validation:** Dengke Wang.

**Writing – original draft:** Xiumei Xu, Dengke Wang.

**Writing – review & editing:** Xiumei Xu.

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
