## [Decision Letter · Decision Letter 0]

11 Dec 2025

Dear Dr. XU,

Thank you for submitting your manuscript to PLOS ONE. After careful consideration, we feel that it has merit but does not fully meet PLOS ONE’s publication criteria as it currently stands. Therefore, we invite you to submit a revised version of the manuscript that addresses the points raised during the review process.

We look forward to receiving your revised manuscript.

Kind regards,

Umberto Baresi, Ph.D.

Academic Editor

PLOS ONE

Journal Requirements:

“This work is supported by the Yunnan Province Philosophy and Social Science Planning (No. ZX2024ZD10).”

“This work is supported by the Yunnan Province Philosophy and Social Science Planning (No. ZX2024ZD10).”

“This work is supported by the Yunnan Province Philosophy and Social Science Planning (No. ZX2024ZD10).”

5. We note that Figure 1 in your submission contain map images which may be copyrighted. All PLOS content is published under the Creative Commons Attribution License (CC BY 4.0), which means that the manuscript, images, and Supporting Information files will be freely available online, and any third party is permitted to access, download, copy, distribute, and use these materials in any way, even commercially, with proper attribution. For these reasons, we cannot publish previously copyrighted maps or satellite images created using proprietary data, such as Google software (Google Maps, Street View, and Earth). For more information, see our copyright guidelines: http://journals.plos.org/plosone/s/licenses-and-copyright.

1. You may seek permission from the original copyright holder of Figure(s) [#] to publish the content specifically under the CC BY 4.0 license.

Additional Editor Comments:

Dear Authors,

I would like to thank you for submitting your work for publication in PLOS ONE.

After this first round of reviews, I am pleased to convey that your work has been deemed as suitable for publication. The anonymous reviewers indicated that for your work to meet its full potential, it requires major restructuring in theoretical framing, literature review, conceptual model design, and methodological transparency.

Specifically, "the conceptual foundations and operationalization require substantial revision".

Please consider also the attached document, in which additional comments from Reviewer #2 are provided.

Please consider all attached feedback before submitting the edited version of your work.

As a requirement for resubmission, please attach the edited version of the manuscript with Changes Tracked in a different color, and a table in which each comment in listed in rows, for which I would ask you to indicate how and where in the document the edits took place.

Thank you.

Reviewers' comments:

Reviewer's Responses to Questions

**Comments to the Author**

1. Is the manuscript technically sound, and do the data support the conclusions?

Reviewer #1: Partly

Reviewer #2: Partly

2. Has the statistical analysis been performed appropriately and rigorously?

Reviewer #1: Yes

Reviewer #2: Yes

3. Have the authors made all data underlying the findings in their manuscript fully available?

Reviewer #1: Yes

Reviewer #2: Yes

4. Is the manuscript presented in an intelligible fashion and written in standard English?

Reviewer #1: No

Reviewer #2: Yes

Reviewer #1: Thank you for the opportunity to review this manuscript. The topic of ecological co-management in agricultural regions is important, and integrating TPB with social network embeddedness has potential value. However, several substantial issues need to be addressed before the manuscript can be considered further. Below I present key concerns, each followed immediately by a recommendation to improve clarity, theoretical rigor, and methodological soundness.

1. Introduction and Problem Definition

The introduction outlines global ecological issues but does not clearly explain the specific ecological challenges in the Lujiangba coffee region or why ecological co-management is necessary. This limits the contextual relevance of the study.

Recommendation: Strengthen the introduction by describing concrete local ecological problems (e.g., soil erosion, chemical overuse, biodiversity decline) and clarifying why this region is an appropriate case.

2. Literature Review and Research Gap

The literature review summarizes TPB and SNET but does not develop a coherent argument leading to a clear research gap. The integration of the two theories is not justified, and existing studies combining networks with behavioral intention are not adequately discussed.

Recommendation: Reorganize the review to (a) describe previous studies on TPB in environmental participation, (b) show how relational embeddedness influences cooperation, and (c) highlight the gap this study addresses.

3. Hypotheses Development and Conceptual Model

Many hypotheses are stated without adequate theoretical justification, and the conceptual model is difficult to interpret. Additionally, constructs such as PO (Perception of Outcome) appear to be used as substitutes for established TPB components, which may lead to conceptual confusion.

Recommendation: Move the hypotheses section directly after the literature review, provide theory-based reasoning for each hypothesis, and revise the conceptual diagram for clarity. If PO is intended to function as a determinant of intention, the authors should explicitly justify this adaptation of TPB using previous studies and clearly explain how PO relates to attitude or intention in this context.

4. Methodology and Sampling

The manuscript does not clearly describe the sampling method, recruitment procedure, or how the participants represent “multiple subjects.” Based on the demographic distribution, the sample appears to consist mostly of residents rather than diverse stakeholder groups. This undermines the claim of multi-subject co-management.

Recommendation: Provide explicit details about the sampling strategy and justify how the selected respondents represent the required actors (e.g., farmers, local officials, business owners). Clarify ethical approval and informed consent procedures, which are required for PLOS ONE. If the study targets only a subset of stakeholders, define the study population accordingly.

5. Measurement Scales and Variable Operationalization

Some items do not align with validated TPB or SNET measurement scales. A few items appear leading or double-barreled, and the distinction between constructs such as ENW, SNW, and behavioral antecedents is sometimes unclear.

Recommendation: Revise the measurement scales to align with validated instruments or justify deviations. Modify vague or leading items.

6. Results Interpretation

Although the statistical procedures are appropriate, the interpretation of mediated effects and relationships lacks theoretical depth. The distinction between direct and indirect pathways is described but not meaningfully explained.

Recommendation: Expand the explanation of why ENW might influence subjective norms more strongly and why SNW may relate more closely to perceived behavioral control. Connect these interpretations to previous environmental psychology and network studies to strengthen the theoretical contribution.

7. Discussion, Policy Implications, and Limitations

The discussion mainly repeats the results rather than offering deeper theoretical or practical insights. Policy recommendations are broad and not clearly derived from the findings. Additionally, limitations regarding sample size, generalizability, and measurement issues are not acknowledged.

Recommendation: Strengthen the discussion by explicitly linking findings to TPB and SNET, provide evidence-based policy recommendations relevant to ecological co-management in agricultural communities, and include a dedicated limitations section.

8. Abstract

The abstract is too general and does not clearly communicate the context, theoretical approach, methodology, or major findings.

Recommendation: Improve the abstract by briefly stating (a) the ecological problem in the region, (b) the theoretical model (TPB + SNET) and its rationale, (c) sample size and analysis method, and (d) the key direct/indirect effects. A clear statement of contribution should be added to the final sentence.

9. Language and Presentation

The manuscript requires substantial editing for grammar, clarity, and readability. Several sentences are unclear or repetitive. Figures and demographic tables should be simplified or moved to an appendix.

Recommendation: A thorough English language revision is recommended before resubmission.

Reviewer #2: The review is uploaded as an attachment, for improved legibility.

In summary:

- statistical significance analysis does not account for multiple hypothesis testing and associated alpha inflation

(I am hesitant about Question 2 above, but this is not a fatal flaw);

- some conclusions or statements could be reformulated to avoid implying definitive empirical inference about causality,

or to clearly separate empirical results from theoretical interpretation or discussion,

or to avoid statements about comparisons that were not validated statistically;

- additional technical details are necessary to allow better understanding and reproduction of the results.

All these issues can be corrected and I do not see other obstacles to publication; the study is conducted thoroughly and well-written up.

**Do you want your identity to be public for this peer review?** For information about this choice, including consent withdrawal, please see our Privacy Policy

Reviewer #1: **Yes:** Muhammad Waleed Ayub Ghouri

Reviewer #2: No

---

## [Author Response · Author response to Decision Letter 1]

28 Jan 2026

Dear Editor,

We sincerely appreciate your valuable comments on our manuscript and are pleased to be informed that the manuscript is deemed to basically meet the publication requirements after the first round of review. We have carefully read and thoroughly reflected on all the suggestions put forward by the reviewers, particularly those regarding the need for significant adjustments to the theoretical framework, literature review, conceptual model design, and methodological transparency, as well as the requirement for substantial revisions to the conceptual basis and operationalization.

In accordance with your and the reviewers' recommendations, we have conducted a systematic and comprehensive revision of the manuscript. The key revisions are as follows:

1.Reorganized the theoretical framework to strengthen the integration logic of the Theory of Planned Behavior (TPB) and Social Network Embeddedness Theory (SNET);

2. Reconstructed the literature review section to highlight the evolution of relevant research and the positioning of this study;

3. Optimized the conceptual model and clearly explained the operational definitions and measurement methods of each construct;

4. Supplemented methodological details to enhance the transparency and reproducibility of the analysis process.

5. All revisions have been marked in different colors to clearly show the content and location of modifications;

6. Detailedly listing each reviewer's comment, along with explanations of how we addressed them in the manuscript and the specific locations.

We appreciate the time and professional advice you and the reviewers have invested in improving the quality of this manuscript. We believe that this revision has significantly enhanced the theoretical rigor, structural clarity, and methodological completeness of the paper, making it more consistent with the publication standards of your journal. We are happy to make further improvements if additional suggestions are provided.

Best wishes,

Yours sincerely,

Xiumei XU, Ph.D.

---

## [Editor Report · Decision Letter 1]

5 Feb 2026

The behavioral driving mechanism of ecological co-management among multiple subjects from the perspective of social network embedding: evidence from coffee-producing areas in China

PONE-D-25-25310R1

Dear Dr. XU,

We’re pleased to inform you that your manuscript has been judged scientifically suitable for publication and will be formally accepted for publication once it meets all outstanding technical requirements.

Kind regards,

Umberto Baresi, Ph.D.

Academic Editor

PLOS One

Additional Editor Comments (optional):

Dear Authors,

I believe that you have done an excellent work in addressing the reviewers' comments.

As a result, the manuscript has improved in content and in clarity.

I am glad to recommend this paper for publication.

Kind regards
---

## [Editor Report · Acceptance letter]

PONE-D-25-25310R1

PLOS One

Dear Dr. XU,

I'm pleased to inform you that your manuscript has been deemed suitable for publication in PLOS One. Congratulations! Your manuscript is now being handed over to our production team.

Kind regards,

on behalf of

Dr. Umberto Baresi

Academic Editor

PLOS One